# Assessment of Aspartame (E951) Occurrence in Selected Foods and Beverages on the German Market 2000–2022

**DOI:** 10.3390/foods12112156

**Published:** 2023-05-26

**Authors:** Sydney Schorb, Katharina Gleiss, Roland Wedekind, Eero Suonio, Ann-Kathrin Kull, Marcel Kuntz, Stephan G. Walch, Dirk W. Lachenmeier

**Affiliations:** 1Chemisches und Veterinäruntersuchungsamt (CVUA) Karlsruhe, Weissenburger Strasse 3, 76187 Karlsruhe, Germany; sydney.schorb@cvuaka.bwl.de (S.S.); katharina.gleiss@cvuaka.bwl.de (K.G.); marcel.kuntz@cvuaka.bwl.de (M.K.); stephan.walch@cvuaka.bwl.de (S.G.W.); 2Evidence Synthesis and Classification Branch, International Agency for Research on Cancer, 25 Avenue Tony Garnier, 69366 Lyon, France; wedekindr@iarc.who.int (R.W.); suonioe@iarc.who.int (E.S.)

**Keywords:** aspartame, sweeteners, food additives, food control, German national survey, food safety, risk assessment

## Abstract

This study examines the occurrence of the artificial sweetener aspartame (E951) in foods and beverages sampled by food control authorities in Germany between 2000 and 2022. The dataset was obtained through the Consumer Information Act. Out of 53,116 samples analyzed, aspartame was present in 7331 samples (14%), of which 5703 samples (11%) in nine major food groups were further evaluated. The results showed that aspartame was most commonly found in powdered drink bases (84%), flavored milk drinks (78%), chewing gum (77%), and diet soft drinks (72%). In the solid food groups, the highest mean aspartame content was detected in chewing gum (1543 mg/kg, *n* = 241), followed by sports foods (1453 mg/kg, *n* = 125), fiber supplements (1248 mg/kg, *n* = 11), powdered drink bases (1068 mg/kg, *n* = 162), and candies (437 mg/kg, *n* = 339). Liquid products generally had the highest aspartame content in diet soft drinks (91 mg/L, *n* = 2021), followed by regular soft drinks (59 mg/L, *n* = 574), flavored milk drinks (48 mg/kg, *n* = 207), and mixed beer drinks (24 mg/L, *n* = 40). These results suggest that aspartame is commonly used in some foods and beverages in Germany. The levels of aspartame found were generally within the legal limits set by the European Union. These findings provide the first comprehensive overview of aspartame in the German food market and may be particularly useful in informing the forthcoming working groups of the WHO International Agency for Research on Cancer (IARC) and the WHO/FAO Joint Expert Committee on Food Additives (JECFA), which are in the process of evaluating the human health hazards and risks associated with the consumption of aspartame.

## 1. Introduction

Aspartame (E951, CAS# 22839-47-0) is a commonly used artificial sweetener in low-calorie beverages, prepared foods, and tabletop sweeteners that was approved for use in over 90 countries since the 1980s, and concerns about its safety and potential health risks have led to calls for further information and research. In 1981, the WHO/FAO Joint Expert Committee on Food Additives (JECFA), as part of a program to assess the risks of food additives and chemicals, evaluated its health effects and established an acceptable daily intake for aspartame of 0–40 mg/kg bw [1]. However, the WHO International Agency for Research on Cancer (IARC) Monographs programme, which identifies potential carcinogenic hazards, has not evaluated aspartame to date. A recent report of the IARC priorities advisory group [2,3] notes that while previous epidemiologic studies did not typically find an association between aspartame intake and cancer risk, some new studies have suggested the possibility of a link. Among them, a prospective study of hematologic malignancies found a positive association with multiple myeloma and non-Hodgkin lymphoma in men [4]. Moreover, a case-control study of exocrine pancreatic adenocarcinoma found a positive association with low-calorie soft drink consumption in men [5], and another case-control study showed a positive association between regular use of artificial sweeteners and urinary tract tumors [6]. More recently, a large cohort study of 102,865 French adults showed that aspartame intake was associated with an increased risk of breast and obesity-related cancers [7]. Concerns have also been raised by a few studies in experimental animals showing dose-related increased risks of certain cancers at exposure levels previously considered safe for human consumption [8]. Mechanistic studies relevant to the key characteristics of carcinogens have also been carried out (e.g., [9,10,11]). For these reasons, the IARC priorities advisory group recommended aspartame as a high priority for evaluation [2,3]. In addition to IARC, JECFA has also recommended aspartame as a high-priority substance for re-evaluation [12].

According to the proposals, both the IARC Monographs programme and the JECFA will conduct complementary evaluations in 2023: IARC will investigate whether aspartame has any potential carcinogenic effects (hazard identification), while JECFA will update its risk assessment by reviewing the acceptable daily intake and dietary exposure assessment of aspartame [13]. The sequence of these evaluations will allow for a comprehensive evaluation of the health effects of aspartame consumption based on the latest available evidence.

Both IARC and JECFA have published calls for data [13,14] that include data relevant to the dietary exposure assessment, such as the level of use of the additive in foods. To properly characterize the exposure of the general population and specific subgroups to aspartame, detailed data on its content in a variety of food groups are needed. However, although aspartame is one of the most commonly used artificial sweeteners, there is a paucity of data in the scientific literature on its occurrence in various foods other than artificially sweetened beverages. To provide the agencies with such data from Germany to inform their exposure assessments, this article examines the results of a recent request for information on aspartame, providing a summary of the data and an analysis of their implications.

## 2. Materials and Methods

### 2.1. Data Request under the Consumer Information Act

The official food control laboratories in Germany collected and analyzed the samples in this dataset between 2000 and 2022. Analytical methods were validated and externally accredited as part of government food control efforts according to the guidelines of ISO/IEC 17025 [15]. The methodology for sample selection in the study is a combination of convenience sampling and systematic testing, such as annual food monitoring projects. Samples were collected using a risk-based sampling approach, which means that samples are not selected at random, but based on a perceived risk of contamination or non-compliance [16]. Samples were collected in all German states from a variety of sources, including retailers, importers, and manufacturers, which can introduce some bias into the results, such as the focus of food control on the “bottleneck”, i.e., primarily at the level of the manufacturers or importers, rather than at the level of sale to the end consumer. Sampling bias could therefore include a lack of inclusion of possible aspartame degradation during the shelf life of the food. The data did not include information on the consumption patterns of the products, which limits the ability to estimate actual exposure levels.

In response to a consumer request from the last author (D.W.L.), the German Federal Office of Consumer Protection and Food Safety (BVL) collected and consolidated the data used in this study, which was published on the FragDenStaat.de website in March 2023 [17]. The request for the data was made under the Consumer Information Act, which requires food control authorities to provide information to consumers on food, feed, consumer products, and cosmetics. This includes information on unauthorized deviations from requirements, as well as measures and decisions taken in response to deviations and potential health and safety risks posed by products. The act also covers monitoring activities and measures for consumer protection [18].

Access to the complete dataset, as well as the original consumer request, can be found on the internet portal FragDenStaat.de, which is managed by the non-profit association Open Knowledge Foundation Deutschland e.V. (located in Berlin, Germany) [17].

### 2.2. Data Description and Analysis Methods

The available data include information such as the year and type of store where the sample was collected, the country of origin, the product group and matrix of the sample, the aspartame content, the analytical methods, and the limit of detection (LOD) and the limit of quantification (LOQ) of the method used. In total, this dataset contained 53,116 analytical results for aspartame in different food products. The dataset included separate tables with qualitative data (*n* = 720) and quantitative data (*n* = 52,396). In this study, only the quantitative data were further evaluated.

The analytical methods used to determine the aspartame content in the major food groups were primarily test methods according to the official collection of paragraph 64 of the German Food, Commodities and Feed Code (Lebensmittel-, Bedarfsgegenstände- und Futtermittelgesetzbuch) for food control authorities and testing institutions. This ensures consistent quality of testing and comparability of results [19]. Generally, the high-performance liquid chromatography (HPLC) method BVL L 00.00-28 is used, which is an adoption of the DIN EN 12856 standard with identical wording [20]. Some laboratories have used other validated and accredited methods, based on HPLC coupled with different detectors, or based on nuclear magnetic resonance (NMR) spectroscopy (e.g., [21]). The method used for each analytical result is indicated in the raw data Excel table called “Anlage 5” of the dataset [17].

### 2.3. Data Analysis and Selection of Food Items for Inclusion: Regrouping and Prioritizing Foods

Analysis of the dataset presented in this study, including all statistical calculations, was performed using Microsoft Excel version 2016 (Microsoft, Redmond, WA, USA).

During the evaluation of the data, it quickly became apparent that most samples (86% of the total sample, 87% of the quantitatively analyzed samples) were negative for aspartame, i.e., the food was analyzed but aspartame was not detected. This finding can be explained by the fact that multiparameter methods, which analyze several sweeteners with the same assay, are commonly used for food control. This means that aspartame could have been measured even if the intention was to measure another sweetener. Therefore, it is important to note that the dataset is heavily biased toward foods that were suspected to contain aspartame or other compounds that were measured with the same assay. This implies that the sample of foods in Germany is not representative and that the mean values should be interpreted with caution.

The original BVL grouping of the raw data was less informative (see the Excel table called “Anlage 7” of the original dataset [17]) because only some subgroups of the major food groups appeared to contain aspartame. For example, in the dairy category, aspartame was found only in certain subgroups such as ready-to-drink buttermilk beverages. It was decided to re-analyze the data and categorize them into more meaningful groups using the standardized FoodEx2 food classification and description system of the European Food Safety Authority (EFSA) [22]. Only food groups in which at least 20 samples were analyzed and at least 40% of the samples were found to contain aspartame were generally retained in the analysis. Exceptionally, fiber supplements were included despite only 13 samples being analyzed, but with a comparably high frequency of positives (55%).

### 2.4. Application of Artificial Intelligence Tools

Manuscript editing, data textualization and summarization, and German-English translation were performed using ChatGPT, an artificial intelligence (AI) language model developed by OpenAI (https://chat.openai.com/). ChatGPT provided helpful insights and suggestions throughout the writing process, and its advanced language processing capabilities improved the clarity and accuracy of the manuscript. Specifically, the literature data in the first part of the discussion section, which were only available as tabulated data in governmental reports, were textualized using ChatGPT. The final manuscript was edited using DeepL Write (https://www.deepl.com/write) and Trinka (https://www.trinka.ai/) to improve the English language and grammar. These tools utilize AI technology to suggest text changes, suggest synonyms, and improve overall writing quality. Trinka.ai also included publication readiness checks, such as checks for technical compliance, reference validation, figure and table order, keywords, and summary suggestions. These tools did not contribute intellectually to the writing process, and the authors are fully responsible for the originality, validity, and integrity of the content of this manuscript.

## 3. Results

Out of a total of 53,116 samples included in the dataset, 7331 samples (14%) contained aspartame. Based on the selection criteria of minimum detection frequency and number of samples, 5703 samples (11%) in nine major food groups were selected for further evaluation. Samples not belonging to the selected food groups, 1628 samples (3%), were excluded.

Food items were grouped according to their respective FoodEx2 product groups (flavored milk drinks, soft drinks, diet soft drinks, mixed beer drinks, candies, chewing gum, powdered drink bases, sports foods, and fiber supplements). The product groups and the corresponding individual products included in each product group are listed in Table A1 in Appendix B. Table 1 shows these food groups with the calculated percentage of positive samples.

Most of the samples were collected from German retailers. Additionally, samples were collected from German kiosks, restaurants, grocery stores, drugstores, canteens, breweries, butchers, bakeries, fitness centers, and all other food outlets or directly from importers. According to the labeling, the main origin of the samples’ manufacturer was Germany (79%). A high number of samples did not have information on the country of origin (9.5%). If known, the next most commonly occurring countries were Poland (1.0%), The Netherlands (0.8%), France (0.4%), Turkey (0.4%), Denmark (0.3%), China (0.2%), Belgium (0.2%), and the United Kingdom (0.2%) (Appendix A, Appendix A).

The main descriptive statistical parameters, including mean, standard deviation, median, maximum, and 95th percentiles, were calculated from the positive samples and are presented in Table 2 (Appendix A, Appendix A). The highest mean levels of aspartame were found in sports foods and chewing gum. The data also showed that the levels of aspartame varied considerably within food categories, with some products containing much higher levels than others.

The data analysis did not reveal any obvious trends or changes in content over time (Appendix B, Figure A1), even when compared to the previous literature. Due to incomplete data for some years, which varied by product group, it was not possible to perform a meaningful analysis of changes or trends in aspartame concentrations over time.

Finally, the data were re-evaluated against the EU maximum levels. Generally, the levels of aspartame found were within the legal limits. Apart from the isolated outliers, there were only two exceedances (diet soft drinks: 970 mg/L, 636 mg/L) which were not excluded. The isolated outliers (see footnotes in Table 2) were considered to be data entry errors due to their technologically unlikely high levels.

## 4. Discussion

According to market data published by the German Federal Institute for Risk Assessment, between 2015 and 2018, 989 foods and 1055 beverages with added non-nutritive sweeteners (NNS) were launched in Germany, of which 16% contained aspartame [24]. Despite this common use, a lack of occurrence and exposure data was noted in Germany. The last major European assessment by EFSA in 2013 [25] did not include data from Germany.

The only comprehensive study of aspartame exposure in Germany was published by Bär and Biermann in 1992 [26]. In this study, the occurrence data of aspartame in different food and beverage products were evaluated by different methods. First, the researchers collected data from food manufacturers by requesting information on the sweetener content of their products. Second, package labels were examined to determine the presence and amount of aspartame. Finally, chemical analysis was used in cases where information was not readily available. These multiple complementary approaches provided a comprehensive understanding of the prevalence of aspartame in the marketplace and allowed for a comprehensive exposure assessment when the data were combined with food frequency questionnaires, but the authors did not publish the occurrence data so no comparison with this survey is possible. Currently, quantitative labeling of aspartame is not required, so only the qualitative presence of the compound is indicated on the ingredient list. Therefore, it is not possible to assess quantitative occurrence by evaluating package labels.

Some data from previous surveys have been published, mostly in government reports. As part of the German National Surveillance Plan 2006, fruit, vegetable, and mushroom products were analyzed for additives. Aspartame was not detected in any of the 237 samples analyzed [27]. According to the German National Surveillance Plan 2007, the occurrence of aspartame was determined in different types of beverages. Of the 170 samples of fruit juice drink samples analyzed, 29 contained aspartame at levels ranging from 11 to 381 mg/kg, with a mean of 85 mg/kg. Of the 51 nectar samples tested, only one was positive for aspartame at 0.1 mg/kg. In other types of beverages, 21 of 124 samples were positive for aspartame, with levels ranging from 18 to 444 mg/kg and an average of 115 mg/kg. None of the samples tested in any category exceeded the legal limit [28]. According to the 2008 German National Surveillance Plan, sweeteners in confectionery without added sugar (including hard and soft candies and confectionery for diabetics) were analyzed. Of a total of 353 samples, 185 contained aspartame. The mean amount of aspartame was 404 mg/kg, with a maximum of 1203 mg/kg. The 90th percentile value was 774 mg/kg. The maximum limit for aspartame of 1000 mg/kg was exceeded in one sample, which was an unfilled hard candy [29]. During the German National Food Monitoring 2015, mineral waters, including raw water, were analyzed for selected sweeteners. Aspartame was not detected in any of the 19 samples [30].

Van Vliet et al. reported aspartame analyses of seven soft drinks from Germany. Aspartame was detected in five samples with a mean of 119 mg/L (range 53–330 mg/L) [31]. Maes et al. detected aspartame in two brands of diet coke at 138 and 149 mg/L [21].

Recently, the German Federal Institute for Risk Assessment (BfR) published data from an analytical survey of 92 energy-reduced or sugar-free non-alcoholic beverages. For energy-reduced beverages, the study found that in three aspartame-positive samples, the average level of aspartame was 20 mg/kg, with a range of 0.05–45 mg/kg. For sugar-free non-alcoholic beverages, the study found 64 positive samples and found that the average level of aspartame was 75 mg/kg, with a range of 11–492 mg/kg [32].

The results of this study confirm that aspartame is still commonly used in foods in Germany. Given the variation in aspartame content in foods and beverages, the survey data in this study are in reasonable agreement with the results of previous studies. However, this study is based on a much larger number of samples and therefore provides more stable statistics.

The survey of the occurrence of aspartame in Germany based on official food controls has some limitations. The data may not be representative of the general population’s consumption of foods containing aspartame, because the sampling was a combination of systematic and convenience sampling, and a threshold of at least 20 samples was applied in this analysis (except for fiber supplements, *n* = 13). However, the sample size is relatively large, which may indicate that representativeness was achieved for the food groups containing aspartame. This means that the data may still be suitable for exposure analysis of consumers of food groups containing aspartame, even if they are not fully representative of the complete range of food and beverage products available in the German market. Although the large sample size provides a robust representation of aspartame occurrence in selected food and beverage categories, it is important to consider other factors that may influence aspartame consumption. For instance, individual preferences and purchasing habits, including regular consumption of a certain product containing aspartame, may impact the overall exposure to the sweetener. Therefore, the data should be interpreted with caution and further research, such as combining occurrence data with dietary survey data, may be needed to fully understand the extent of aspartame consumption in the population. Second, the samples were only collected in Germany and may not be representative of other countries or regions. Studies reporting the occurrence of aspartame in foods and beverages in other countries are mainly from Europe and focus primarily on beverages. Comprehensive data on the occurrence of aspartame are available from only four European countries [25].

Given this lack of comprehensive data, the main purpose of obtaining the data through the Consumer Information Act was to make the German data on aspartame publicly available so that the conditions of the IARC preamble are met, which require that the data be made publicly available to meet transparency requirements [33]. It is hoped that the data can now be used in the work of the IARC and JECFA working groups. Since this study is the first to provide significant data on food groups such as dietary fiber supplements, certain dairy products, and powders used to make beverages, it is considered critical to providing a comprehensive understanding of aspartame exposure, despite the limitations noted above.

## 5. Conclusions

These data on the occurrence and levels of aspartame in various foods in Germany provide new and updated information on the use of aspartame in the food industry that may help researchers better understand its prevalence and potential health effects. Based on the data provided, it appears that aspartame is present at varying levels in a variety of foods. In terms of the detected levels of aspartame, the data show that the highest concentrations were found in sports foods and chewing gum. The levels of aspartame found were generally within the legal limits set by the European Union.

Consumers should read food labels carefully if they are concerned about their intake of this additive. Although the levels found in this dataset do not exceed the regulatory limits that are currently considered safe [25], this does not necessarily reflect the total aspartame intake of an individual who consumes several foods containing the sweetener.

However, these new data can contribute to a better understanding of the presence and levels of aspartame in different foods and help ensure the safety and health of consumers, especially when considering the cumulative exposure from different food groups, such as sports foods and some dietary supplements, which may have been underestimated.

## Figures and Tables

**Table 1 foods-12-02156-t001:** Overview of the different product groups analyzed for the presence of aspartame.

Product Group	Total Number of Samples	Frequency of Aspartame-Positive Samples [%]
Diet soft drinks	2783	72.7
Soft drinks	1167	49.2
Candies	603	56.2
Chewing gum	312	77.2
Sports foods (with protein and amino acids)	297	42.1
Flavored milk drinks	268	78.0
Powdered drink bases	195	83.6
Mixed beer drinks	58	69.0
Fiber supplements	20	55.0

**Table 2 foods-12-02156-t002:** Aspartame content in various product groups analyzed from 2000 to 2022.

Product Group	Number of Quantifiable Samples	Unit	Mean	Median	Maximum	Standard Deviation	95th Percentile	EU Maximum Level [23]
Chewing gum	241	mg/kg	1543	1369	4617	1042	3649	2500 ^d^ 5500 ^e^
Sports foods (with protein and amino acids)	125	mg/kg	1453	1030	6615	1461	5002	2000 ^f^ 5000 ^f,g^
Fiber supplements	11	mg/kg	1248	1276	1469	175	1460	2000 ^f^ 5000 ^f,g^
Powdered drink bases ^a^	162	mg/kg	1068	1133	4861	672	1600	600 ^h^
Candies	339	mg/kg	473	440	3096	332	890	2000 ^i^ 6000 ^j^
Diet soft drinks	2021	mg/L	91	60	970	101	335	600
Soft drinks ^b^	574	mg/L	59	34	531	74	203	600
Flavored milk drinks ^c^	207	mg/kg	48	47	90	17	79	1000
Mixed beer drinks	40	mg/L	24	26	55	15	42	600

^a^ excluding one data outlier (above 20,000 mg/kg); ^b^ excluding three data outliers (above 20,000 mg/kg); ^c^ excluding two data outliers (above 900 mg/kg); ^d^ with added sugars or polyols; ^e^ with no added sugars; ^f^ maximum levels refer to products ready for consumption, prepared according to the manufacturer’s instructions; ^g^ in chewable form; ^h^ maximum level for flavored drinks in liquid form (the maximum levels refer to the finished products prepared according to the instructions for use provided by the manufacturer and not to the powder bases); ^i^ confectionery, energy-reduced or with no added sugars; ^j^ breath-freshening micro-sweets, with no added sugars.

## Data Availability

Publicly available datasets were analyzed in this study. These data can be found here: https://fragdenstaat.de/en/request/aspartam/ (accessed on 28 March 2023). The data were provided by the German authority Federal Office of Consumer Protection and Food Safety (BVL) in the area of food safety and consumer protection, survey on aspartame, 2000–2022. The own derivative calculations presented in this study are available in the Appendix A.

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
