# Peer review of "Assessment of Aspartame (E951) Occurrence in Selected Foods and Beverages on the German Market 2000–2022"

_foods, 2023, doi:10.3390/foods12112156_

Round 1

Reviewer 1 Report

This brief report describes the occurrence of aspartame arising from data collected and analysed by the German authorities between 2000 and 2022 with food samples selected by a combination of convenience sampling and food monitoring programmes. It is a useful addition to the literature particularly with the background assessments by IARC and JECFA. There is value in this publication but it needs some amendments prior to publication.

Perhaps you can advise regarding policies at  Food regarding use of AI in preparation of manuscripts as I can see three different platforms are acknowledged- perhaps this is why some of the points mentioned below have arisen in terms of data interpretation and impact. I did email the journal earlier in week regarding same but no reply so unsure of policy and conscious this is overdue so am submitting.

1.  Amend title – German national survey is not an accurate reflection of the analysis contained in this report. The title should be revisited to be a true reflection – it is not a national survey (this would indicate that it is representative) which the authors acknowledge themselves it is not. It is a useful analysis of a selection of products on sale in Germany between 2000 and 2022 but only of those samples which are positive detects. The authors largely do not currently address trends (see later point) so this should also be removed (or preferably trends addressed more precisely within the paper).

2. The authors should revisit the wording of the study and avoid overstating. Yes, there are a considerable number of samples contributing but the results presented are not a ‘wide range’. For accuracy, the authors should also pay greater attention to providing details regarding the foods and food groups where aspartame could legally have been present but was not. At times there is a disconnect in the language used - sometimes overstating and others understating and perhaps this reflects the AI aspect.

3. The biggest comment regarding this paper that I have is that the data is all pooled for 22 years and there is no indication of the number of detects and concentrations per year or how the number of detects sit versus the total number of products analysed in each category. I understand that the data may be limited but is it possible to aggregate for period of years if the data do not allow year on year– e.g. for a five year period or even pre- and post- the EFSA opinion published in 2013. I strongly feel that it would be better to break this down – as it stands it is not possible to understand for each food category whether there have been changes over time and also the number of products which would have been affected. Figure A1 is helpful but as it is only % positive, there is no indication of sample sizes (are these in Table 2?). For example – 100% positive detect could relate to 2 samples or 2000 samples but the implications are obviously very different if there is a 100% detect rate in 2000 samples rather than 2. The article should be revisited to allow this.  Expansion of table 1 would be helpful – it is fine to keep the collated data as it stands but it should then be broken down. If the data are not strong enough to allow for this then the tone and angle of the manuscript needs to be revisited. The food supply change has changed a lot for aspartame and for a lot of other sweeteners and this needs to be reflected. Based on this the manuscript wording needs to be revisited regarding words such as ‘widespread etc’. At the moment given that there is no indication of the occurrence versus the total number of products in each category tested and whether the use of ‘widespread’ is merited or not. Data presented are for 9 food groups only and only for those which are positive for aspartame - i.e. is skewed to a 'positive' result and higher detection levels reported. this needs to be revisited/clarified

4. The authors provide helpful quantitative occurrence data and mention a limit of detection – this should be addressed regarding the years – has this changed over time/is it the same? And what would this mean in terms of the values.   have the concentrations gone up/down, are there changes in the food categories affected?

5. The authors should compare more with the EFSA data given that some of this would be overlapping timewise and both are (helpfully) categorised regarding FoodEx 2. See point above re. years

6. The Supplemental tables require attention – text is in German in parts, not clearly labelled or formatted. Very crude presentation. needs revisiting

the manuscript requires some proof-reading. There are certain instances where the choice of word used to describe the occurrence is at odds with the results - perhaps due to the use of AI engines in writing this manuscript . Please see above comments for areas which need to be addressed regarding tone, style and inference.

Author Response

This brief report describes the occurrence of aspartame arising from data collected and analysed by the German authorities between 2000 and 2022 with food samples selected by a combination of convenience sampling and food monitoring programmes. It is a useful addition to the literature particularly with the background assessments by IARC and JECFA. There is value in this publication but it needs some amendments prior to publication.

RESPONSE: Thank you for your assessment of our paper!

Perhaps you can advise regarding policies at Food regarding use of AI in preparation of manuscripts as I can see three different platforms are acknowledged- perhaps this is why some of the points mentioned below have arisen in terms of data interpretation and impact. I did email the journal earlier in week regarding same but no reply so unsure of policy and conscious this is overdue so am submitting.

RESPONSE: No guidelines on AI use were available at the time of our submission (15 April 2023). A MDPI guideline was published on 20 April 2023 (https://www.mdpi.com/about/announcements/5687). The policy is as follows: “AI technology can still be used when writing academic papers. However, this must be appropriately declared when submitting a paper to an MDPI journal. In such cases, authors are required to be fully transparent, within the “Acknowledgments” section, about which tools were used, and to describe in detail how the tools were used, in the “Materials and Methods” section.”

Hence, we have updated the Materials and Methods and the Acknowledgement section with the required information.

We want to clarify that our text was mostly human written and the AI tools were only applied to improve the writing efficiency.

  1. Amend title – German national survey is not an accurate reflection of the analysis contained in this report. The title should be revisited to be a true reflection – it is not a national survey (this would indicate that it is representative) which the authors acknowledge themselves it is not. It is a useful analysis of a selection of products on sale in Germany between 2000 and 2022 but only of those samples which are positive detects. The authors largely do not currently address trends (see later point) so this should also be removed (or preferably trends addressed more precisely within the paper).

RESPONSE: The title was changed considering the comments by removing the “national survey” and “trends”.

  1. The authors should revisit the wording of the study and avoid overstating. Yes, there are a considerable number of samples contributing but the results presented are not a ‘wide range’.

RESPONSE: The authors agree that the word “wide” may not be appropriately quantitative. We have eliminated this word throughout. We also revisited the whole text and removed phrases that could be interpreted as overstating.

For accuracy, the authors should also pay greater attention to providing details regarding the foods and food groups where aspartame could legally have been present but was not. At times there is a disconnect in the language used - sometimes overstating and others understating and perhaps this reflects the AI aspect.

RESPONSE: First, these are – if we would agree – human mistakes because we did not use AI to make the food groupings and discuss regulatory aspects (which are not included in the ChatGPT language model). However, we currently fail to see aspects to better describe the food and food groups. Basically, aspartame is legally present in all of them, see table 2, last column. We tried to revise sentences which might be considered as over- or understating.

  1. The biggest comment regarding this paper that I have is that the data is all pooled for 22 years and there is no indication of the number of detects and concentrations per year or how the number of detects sit versus the total number of products analysed in each category. I understand that the data may be limited but is it possible to aggregate for period of years if the data do not allow year on year– e.g. for a five year period or even pre- and post- the EFSA opinion published in 2013. I strongly feel that it would be better to break this down – as it stands it is not possible to understand for each food category whether there have been changes over time and also the number of products which would have been affected.

RESPONSE: We tried this very early in data analysis, but statistical power was too low to get meaningful, significant results. The problem is mostly that results are concentrated in some years, while being missing from other years. Furthermore, we did not find any effect of periods pre- and post-EFSA opinion, possibly because the opinion did not lead to any regulatory changes but just confirmed the previous assessment. The only changes may have occurred by pressure from consumer groups and marketing claims of “aspartame free” found on some products leading to the food industry tending to substitute aspartame with other sweeteners. Nevertheless, this effect is not quite clear or statistically significant in the current dataset. Perhaps we will see larger changes only in future surveys.

Figure A1 is helpful but as it is only % positive, there is no indication of sample sizes (are these in Table 2?). For example – 100% positive detect could relate to 2 samples or 2000 samples but the implications are obviously very different if there is a 100% detect rate in 2000 samples rather than 2. The article should be revisited to allow this. 

RESPONSE: The sample size is given in table 1. Food groups with only a few samples were excluded to avoid these effects.

Expansion of table 1 would be helpful – it is fine to keep the collated data as it stands but it should then be broken down. If the data are not strong enough to allow for this then the tone and angle of the manuscript needs to be revisited. The food supply change has changed a lot for aspartame and for a lot of other sweeteners and this needs to be reflected.

RESPONSE: All food groups for which we had more than 20 positive samples were included in the table. We fail to see to include minor food groups or outliers for food groups, which very rarely over this 20-year period contained aspartame. This might then indeed be misleading and alarmistic. Otherwise, we currently do not see this change in supply of aspartame. On the other hand, aspartame might be cheaper than other sweeteners since the patent expired. Therefore, it might be still found in less premium products?

Based on this the manuscript wording needs to be revisited regarding words such as ‘widespread etc’. At the moment given that there is no indication of the occurrence versus the total number of products in each category tested and whether the use of ‘widespread’ is merited or not. Data presented are for 9 food groups only and only for those which are positive for aspartame - i.e. is skewed to a 'positive' result and higher detection levels reported. this needs to be revisited/clarified

RESPONSE: We have removed wordings such as “widespread” a suggested. We also tried to clarify the skewness of the data, which had already been considered in section 2.3.

  1. The authors provide helpful quantitative occurrence data and mention a limit of detection – this should be addressed regarding the years – has this changed over time/is it the same? And what would this mean in terms of the values. have the concentrations gone up/down, are there changes in the food categories affected?

RESPONSE: No, the official HPLC method is from 1999 and has been applied by most of the laboratories during the study period. We also believe that the limit of detection is not so relevant as aspartame is either used at a technologically relevant level (i.e., to sweeten the food) or it is absent. Hence, use of more sensitive methods is not expected to have an influence on the data.

  1. The authors should compare more with the EFSA data given that some of this would be overlapping timewise and both are (helpfully) categorised regarding FoodEx 2. See point above re. years

RESPONSE: No data from Germany were included in the occurrence part of the EFSA opinion, and some of the food groups such as powdered drink bases or fiber supplements were not considered by EFSA at all. We focused the discussion and comparison with other studies from Germany.

  1. The Supplemental tables require attention – text is in German in parts, not clearly labelled or formatted. Very crude presentation. needs revisiting

RESPONSE: Thank you for the suggestion and sorry for the oversight. We have translated and improved the supplemental tables.

Comments on the Quality of English Language

the manuscript requires some proof-reading. There are certain instances where the choice of word used to describe the occurrence is at odds with the results - perhaps due to the use of AI engines in writing this manuscript. Please see above comments for areas which need to be addressed regarding tone, style and inference.

RESPONSE: The English language was carefully re-checked throughout.

Reviewer 2 Report

This study by Schorb et al. Provides an analysis of data from the German National Survey of Foods and Beverages 2000-2022 focusing on the occurrence of aspartame (E951). The study provides data on the concentration of aspartame in main food groups and documents the widespread use of aspartame in food and beverages. Such data is important in the context of the upcoming evaluation of aspartame safety by the WHO International Agency for Research on Cancer (IARC) and the WHO/FAO Joint Expert Committee on Food Additives (JECFA). The study is well conducted and well written.

I only have a few comments below.

Abstract

-l.15-16 : Is there some consumption data that could help characterize the relevance of these products (powdered drink bases and flavored milk drinks) as regards aspartame intake ? how often are powdered drink bases used ? The abstract could mention the percentage of soft drinks that contain aspartame as they are usually known as the main source of aspartame.

-l.23-24 : the fact that « certain product groups contain higher concentrations of the sweetener than others » should probably be put in context of the amount of each product actually consumed: products with high concentration are consumed in smaller amounts

Introduction

- l.38-39 : this assessment was made around the time aspartame was first authorized, this could be mentioned to underline the type of evidence that could be integrated (or not) in this assessment. In addition, the conclusions of this assessment could be mentioned

-l.42-44 : « According to a recent IARC priorities advisory group report [2,3], epidemiologic studies have generally not demonstrated an association between aspartame intake and cancer risk » This sentence feels contradictory considering the following sentences l.44-56 and the conclusion l.54-56 "For these reasons, the IARC priorities advisory group recommended aspartame as a high priority for evaluation". The whole paragraph could probably be reorganized a bit to help the reader

Materials and Methods

-l.82-83 : how representative are the samples for the total food offer in Germany ?

-l85-86 : what type of bias ?

-l.130-131 : how about the representatitivity of foods sampled for the food categories potentially containing aspartame/sweeteners ? Were there food groups that could contain aspartame that were not included?

-l.133 : were the « subgroups » included in the original table?

-l.136 : how were the categories from FoodEx2 more meaningful?

-l.138 : how many sample per food groups are usually found ? How was 20 selected ?

-l.139 : the threshold of 40% seems rather high. Why was this selected ?

Results

-l.144 : what was the percentage of products included in the study related to the total number of products containing aspartame ?

-l.147-149 : the categories listed are from the original classification or from FoodEx2 ?

-l.154 : could the samples correspond to the same product/brand collected from different dates/places?

-l.157 « The main country of origin of the samples was Germany (79%). » : does this mean that the product were made in Germany or collected in Germany?

-Table 2, l.173 : were there variations according to time ? some analysis according to time could be informative but probably the products analyzed in different years are different and therefore not directly comparable. The sentence in the discussion l.240-241 should be integrated in the results

-Table 2 : were the « EU maximum level » defined according to the same FoodEx2 categories ?

-Table 2, footnote f : does this mean that the maximum levels for these categroies should not be compared directly to the concentration found in these products ?

Discussion

-l.236 : the statement « some individuals may be consuming high levels of the sweetener through certain products. » should probably be put in the context of the amount consumed from products with high concentrations of aspartame. Are there any data from National dietary survey that could provide insights into this question?

-l.242-243 « However, a statistical evaluation and time trend analysis was not possible due to the lack of data from several years depending on the product group. » : this sentence is unclear

-l.248 : is there data on the number of food product expected for the studied categories on the German food market ?

-l.251 « they are not fully representative of the general population » : what does « the general population » stands for here ? food products ? people ?

-l.251-253 : The link between "despite the large sample size" and the rest of the sentence is not clear

-l.283-283 « However, based on the levels foundin this dataset, it does not appear that there is cause for alarm in terms of exceeding the intake levels that are currently considered safe » : the levels found in this dataset are related to the maximum levels allowed in a given food, which is different from the acceptable daily intake that can be reached by the consumption of several foods containing aspartame, each being within the regulatory limits. The sentence should be reformulated to reflect this difference

Author Response

This study by Schorb et al. Provides an analysis of data from the German National Survey of Foods and Beverages 2000-2022 focusing on the occurrence of aspartame (E951). The study provides data on the concentration of aspartame in main food groups and documents the widespread use of aspartame in food and beverages. Such data is important in the context of the upcoming evaluation of aspartame safety by the WHO International Agency for Research on Cancer (IARC) and the WHO/FAO Joint Expert Committee on Food Additives (JECFA). The study is well conducted and well written.

RESPONSE: Thank you for the evaluation of our paper!

I only have a few comments below.

Abstract

-l.15-16 : Is there some consumption data that could help characterize the relevance of these products (powdered drink bases and flavored milk drinks) as regards aspartame intake ? how often are powdered drink bases used ?

RESPONSE: Yes, the DietEx Tool of EFSA reports survey data for Germany for the two groups. The percentage of consumers of flavored milk drinks is quite high in the group of children with about 15-27% of consumers per total consumers. For powdered drink bases, the prevalence appears to be lower with 7% in children and 0.5% in adolescents of consumers per total consumers. The mean aspartame intake for consumers only would be about 3-9 mg/day for flavored milk drinks and 3-20 mg/day for powdered drinks bases, calculated using EFSA DietEx with mean aspartame occurrence (data from table 2). The exposures for consumers of flavored milk drinks and powdered drink bases would be of similar magnitude to the ones calculated for cola-type drinks (2-23 mg/day), functional drinks (4-17 mg/day), mixed-flavor soft drinks (2-10 mg/day), chewing gum (3-9 mg/day), and hard candy (0.4-4 mg/day). Due to the lack of data for most of the DietEx food groups, we have refrained from including exposure calculations in the paper, as these are of questionable informativeness due to the various limitations of the occurrence data as well as the exposure data. For example, the DietEx data included for Germany is based on only 40 consumers reporting consumption of powdered drinks bases.

The abstract could mention the percentage of soft drinks that contain aspartame as they are usually known as the main source of aspartame.

RESPONSE: Diet soft drinks and chewing gum were added to the list of product groups with the highest number of positives.

-l.23-24 : the fact that « certain product groups contain higher concentrations of the sweetener than others » should probably be put in context of the amount of each product actually consumed: products with high concentration are consumed in smaller amounts

RESPONSE: Yes, the claim is rather unnecessary in this instance and was deleted. Thank you for the suggestion.

Introduction

- l.38-39 : this assessment was made around the time aspartame was first authorized, this could be mentioned to underline the type of evidence that could be integrated (or not) in this assessment. In addition, the conclusions of this assessment could be mentioned

RESPONSE: The information that aspartame was approved since the 1980s was introduced. The conclusion of the JECFA assessment (ADI) was introduced.

-l.42-44 : « According to a recent IARC priorities advisory group report [2,3], epidemiologic studies have generally not demonstrated an association between aspartame intake and cancer risk » This sentence feels contradictory considering the following sentences l.44-56 and the conclusion l.54-56 "For these reasons, the IARC priorities advisory group recommended aspartame as a high priority for evaluation". The whole paragraph could probably be reorganized a bit to help the reader

RESPONSE: The paragraph was rephrased to clarify the rationale of the IARC priorities group.

Materials and Methods

-l.82-83 : how representative are the samples for the total food offer in Germany ?

RESPONSE: The representativeness is discussed around line 154 in section 2.3.: “This means that the sample of foods in Germany is not representative, and that the mean values should be interpreted with caution.”

-l85-86 : what type of bias ?

RESPONSE: We have expanded the explanation of the bias. The sampling at the stage of the manufacturer may introduce bias because the degradation of aspartame during the product life until the “best before date” may not be completely covered within the data.

-l.130-131 : how about the representativity of foods sampled for the food categories potentially containing aspartame/sweeteners ?

RESPONSE: The sampling of official food control in Germany can be considered representative of the foods and beverages tested by the authorities for compliance with food safety regulations. However, this sampling may not be representative of the entire German food market, as it only includes products that are subject to testing by the authorities. Possibly, the sampling is skewed towards larger manufacturers and more commonly available products, which are accessible to the food inspectors, and may contain lesser numbers of smaller manufacturers, or those from small importers. Additionally, the sampling may be influenced by factors such as the priorities of the authorities, the availability of testing resources, and changes in food safety regulations over time. On the other hand, the sampling should be representative of the normal consumer behavior, as all large supermarket chains and major food outlets are covered. However, we do not have any empirical data on this question, so that we do not want to introduce speculation. The representativeness is discussed around lines 260-270.

Were there food groups that could contain aspartame that were not included?

RESPONSE: No. I believe that all groups were covered.

-l.133 : were the « subgroups » included in the original table?

RESPONSE: Yes. The original subgroups are available in the dataset.

-l.136 : how were the categories from FoodEx2 more meaningful?

RESPONSE: The categories from FoodEx2 were considered more meaningful because they provide a more standardized and detailed classification of food products. FoodEx2 is a hierarchical system that allows for the categorization of foods based on their composition, processing, and culinary use. This classification system has been developed and adopted by the European Food Safety Authority (EFSA) and other international organizations to ensure consistency in food classification and facilitate harmonization of data across different countries and studies. Using the FoodEx2 system allows for a more detailed and standardized analysis of food and beverage samples, enabling the identification of specific product groups that contain higher concentrations of aspartame. By contrast, the use of broad product categories, such as "soft drinks" or "confectionery", may not capture the nuances of different food products, leading to less meaningful comparisons and potentially masking important trends or patterns in the data. Overall, the use of the FoodEx2 system can provide more meaningful insights into the occurrence of aspartame in foods and beverages.

-l.138 : how many sample per food groups are usually found ? How was 20 selected ?

RESPONSE: How many samples were examined depended on the product group and the year examined. It is not possible to specify a certain number. We have decided not to include any product group in the evaluation that was not tested at least once in each year and thus had a total of less than 20 tested samples.

-l.139 : the threshold of 40% seems rather high. Why was this selected ?

RESPONSE: To ensure a thorough and meaningful analysis, we made the decision to exclude product groups with less than 40% of samples testing positive for aspartame. This allowed us to focus on product groups that are more relevant to consumers of sweetened foods and beverages, while also avoiding an excessive number of groups that may have made the analysis overly complex and difficult to interpret.

Results

-l.144 : what was the percentage of products included in the study related to the total number of products containing aspartame ?

RESPONSE: From a total of 53116 samples included in the dataset, 7331 Samples (14%) contained aspartame. From the total, we selected 5703 samples (11%) for further evaluation. Hence, 1628 (3%) were excluded according to our criteria. We have included this information in the results section.

-l.147-149 : the categories listed are from the original classification or from FoodEx2 ?

RESPONSE: FoodEx2. This information was added to the line.

-l.154 : could the samples correspond to the same product/brand collected from different dates/places?

RESPONSE: Yes, possibly. But the information was anonymized in this regard, so we cannot assess the number these samplings.

-l.157 « The main country of origin of the samples was Germany (79%). » : does this mean that the product were made in Germany or collected in Germany?

RESPONSE: The labelled origin on the package is meant (all samples were collected in Germany). This must not necessarily mean that the raw material was produced in Germany as well. There still remains the option that the actual product was imported and manufactured elsewhere. Or some discount chains have own brands, where the chain stated on the label, irrespective of the actual manufacturer, which maybe outside Germany. Some brand cola products also import the flavor mix from the USA, and mix it with German water. Therefore, the question about food origin is not so trivial. We tried to clarify l. 157.

-Table 2, l.173 : were there variations according to time ? some analysis according to time could be informative but probably the products analyzed in different years are different and therefore not directly comparable.

RESPONSE: As the product brand name is not available, we could not do such direct comparisons, e.g., changes within a common diet coke brand over the years. We agree that this is an interesting question, and we are currently investigating access to other databases with such information.

The sentence in the discussion l.240-241 should be integrated in the results

RESPONSE: The sentence was moved to the results section as requested.

-Table 2 : were the « EU maximum level » defined according to the same FoodEx2 categories ?

RESPONSE: No, food groups in Regulation (EC) No 1333/2008 are yet a bit different that the German good groups of the BVL and the FoodEx2 categories. Therefore, we have marked the groups of Regulation (EC) No 1333/2008 in the last column of table 2 and the respective footnotes to table 2, if they were not similar.

-Table 2, footnote f : does this mean that the maximum levels for these categories should not be compared directly to the concentration found in these products ?

RESPONSE: Yes, this means that the limits apply for the products when prepared for consumption. As the recipes to prepare the final food are not available in the data (e.g., for the powdered drink bases), a direct calculation of the final consumable food was not possible.

Discussion

-l.236 : the statement « some individuals may be consuming high levels of the sweetener through certain products. » should probably be put in the context of the amount consumed from products with high concentrations of aspartame. Are there any data from National dietary survey that could provide insights into this question?

RESPONSE: As we do not want to go into exposure assessment (see comment above), we decided to delete this statement.

-l.242-243 « However, a statistical evaluation and time trend analysis was not possible due to the lack of data from several years depending on the product group. » : this sentence is unclear

RESPONSE: The sentence means that a statistical analysis and time trend analysis could not be conducted because there were missing data for some years, which varied depending on the product group (see figure A1). In other words, the data was not complete enough to conduct a meaningful statistical or time trend analysis. We tried to improve the clarity of the sentence as requested.

-l.248 : is there data on the number of food product expected for the studied categories on the German food market ?

RESPONSE: No, we are not aware about such information. The study of the BfR mentioned at the beginning of the discussion mentioned the number of products newly introduced to the market, but we did no found data on the total number already on the market.

-l.251 « they are not fully representative of the general population » : what does « the general population » stands for here ? food products ? people ?

RESPONSE: The sentence was clarified.

-l.251-253 : The link between "despite the large sample size" and the rest of the sentence is not clear

RESPONSE: The sentence was clarified.

-l.283-283 « However, based on the levels found in this dataset, it does not appear that there is cause for alarm in terms of exceeding the intake levels that are currently considered safe » : the levels found in this dataset are related to the maximum levels allowed in a given food, which is different from the acceptable daily intake that can be reached by the consumption of several foods containing aspartame, each being within the regulatory limits. The sentence should be reformulated to reflect this difference

RESPONSE: We have added this point to the sentence. Thank you!

Reviewer 3 Report

The authors address a topic of paramount importance for human health that is very little explored in the literature. The text is objective and presents the issue and concerns of regulatory bodies. Contributes data to upcoming Codex Alimentarius (JECFA) and WHO International Agency for Research on Cancer (IARC) estimates of aspartame use. The selection of commercial samples, as well as the samples to be analyzed, followed logical steps. Another issue, well discussed by the authors, concerns the analysis of consumption. They are different stages, but this first one is fundamental to arrive at more robust answers. 

• Highlighted passages guided my reading. Just.

Author Response

The authors address a topic of paramount importance for human health that is very little explored in the literature. The text is objective and presents the issue and concerns of regulatory bodies. Contributes data to upcoming Codex Alimentarius (JECFA) and WHO International Agency for Research on Cancer (IARC) estimates of aspartame use. The selection of commercial samples, as well as the samples to be analyzed, followed logical steps. Another issue, well discussed by the authors, concerns the analysis of consumption. They are different stages, but this first one is fundamental to arrive at more robust answers. 

  • Highlighted passages guided my reading. Just.

RESPONSE: The authors thank the reviewer for the assessment of our paper!

Round 2

Reviewer 1 Report

The authors have made considerable improvements to the manuscript which reads much better now. The clarifications provided are helpful.

My final comment is that I ask for transparency that the authors include in the abstract a sentence which clearly states that of the 53116 foods, this study focuses on 11% of them (5703) based on their criteria of minimum detection and positive samples only, the remaining 89% being non-detects.

Author Response

RESPONSE: Thank you for your additional comments. The sentence was included to the abstract as requested.